# Influence of Urban Tree Traits on Their Ecosystem Services: A Literature Review

**Danchen Liang** [1,2] **and Ganlin Huang** [1,2,*]

1    Center for Human-Environment System Sustainability (CHESS), State Key Laboratory of Earth Surface Processes and Resource Ecology (ESPRE), Beijing Normal University, Beijing 100875, China; liangdanchen@mail.bnu.edu.cn

2    School of Natural Resources, Faculty of Geographical Science, Beijing Normal University, Beijing 100875, China

\*    Correspondence: ghuang@bnu.edu.cn; Tel.: +86-18601930480

**Abstract:** Trees in cities are vital to the health of the urban ecosystem and residents' wellbeing. Besides providing habitats, trees provide food via fruits and young leaves, reduce temperature, and enrich landscape aesthetics with spring flowers and autumn foliage. The generation and supply of these ecosystem services are closely related to their traits, such as the size of the canopy and the sparseness of the leaves, which directly affect their ability to shade and cool. Studies focusing on ecosystem services often consider green space as a whole, and some distinguish the difference between trees and grass. Relatively fewer studies examined the influence of tree traits on the supply of ecosystem services. Understanding the association among species, traits, and ecosystem services would be helpful in generating actionable knowledge for urban tree planting. However, these studies are often scattered under different research topics. To this end, we reviewed and summarized studies on the relationship between urban tree species/traits and ecosystem services according to provision, regulation, and cultural service types. Based on 45 publications, we found that leaf area, crown size, diameter at breast height, tree height, and leaf shape may influence various ecosystem services. We presented a preliminary framework of "trait- service" and argued that with the help of such a framework, future research should generate actionable knowledge for practitioners to identify potential tree species for selection according to desired services.

**Keywords:** ecosystem service; urban trees; tree trait; green space

## 1. Introduction

Urban trees provide a variety of ecosystem services in cities either by themselves or as a part of green space. Examples include providing food, mitigating heat, and enhancing aesthetic value. Furthermore, trees offer food to insects and birds via their fruits, seeds, leaves, and nectar, which are crucial for maintaining a healthy ecosystem.

Tree species and certain traits are closely related to the types and extent of their ecosystem services. For example, large leaf size contributes to a higher capacity to mitigate urban heat [1], and colorful foliage enhances ornamental value [2]. Many studies focused on one or two services and factors that may influence them. For example, a study found that ginkgo has a strong capacity for water retention but a weak dust retention effect [3]. Because of the richness in anthocyanins, red-leaved species retain their leaves longer than green deciduous species in the fall, which provides aesthetic value for a longer time [4]. In addition, red-leaved species also survive better in harsh environments such as metal-rich soils [4].

Studies focusing on ecosystem services often consider green space as a whole [5]. While some recognized the difference between trees and grass [6,7], relatively few studies examined the influence of tree traits on the supply of ecosystem service [8] and often scattered under different research topics. Each tree species can be considered a bundle of

traits. These traits decide the types and extents of a variety of ecosystem services along with other factors, such as shading, cooling, noise reduction, and stormwater reduction. For example, a study in Beijing discovered that green spaces with a certain proportion of broadleaf trees were more effective in cooling than pure grasslands and shrublands, whereas conifers were typically less effective than shrubs [6]. Tree species selected based on their performance on one ecosystem service may have undesirable consequences on other types of ecosystem services due to potential trade-offs [9]. In order to generate actionable knowledge for urban tree planting, a comprehensive understanding of the association among species, traits, and ecosystem services is needed. In this context, we reviewed existing literature focusing on the association of urban tree species, traits, and relevant ecosystem services. Our research questions are: (1) What are the main ecosystem services that urban trees can provide? and (2) What three traits influence the provision of these ecosystem services? In the following text, we presented findings on tree species and traits by each type of service. Based on existing literature, this review aims to provide a relatively easy way for researchers to obtain a quick overview of connections among trait-species-service in terms of urban trees and shed light on the direction of future research.

## 2. Materials and Methods

We selected peer-reviewed publications using the Web of Science Core Collection by keywords and filtered articles that focused on associations between tree traits and ecosystem services by reading the abstract; as a result, a total of 45 publications were analyzed. We summarized their findings according to the three types of ecosystem services, namely provision, regulation, and cultural services [10].

## 3. Ecosystem Services Provided by Trees

### 3.1. Provision Service

Some urban trees provide food via fruits and young leaves. Edible tree species include ginkgo, hawthorn, persimmon, and apple trees [11]. Research on the provisioning service of urban trees focused on edible tree species and diversity, food security, and relevant planning and management [12]. A study in New York City, USA, identified 201 tree species that can provide food and many more species that can provide medicine [12], indicating that there are a great number of edible tree species in urban environments.

Food provision was essential during economic hardship or for people living in straitened circumstances. There have been historical examples where fruit trees were planted in urban public spaces to provide complementary food. For example, East Berlin had more fruit trees than West Berlin during the Cold War [13]. Similarly, orchards were reported to support the lives of residents in poverty during the early industrialization phase in the UK, Germany, and Sweden [11,14].

Nowadays, initiatives promote community orchards to enhance food security and provide fresh produce to low-income people. For example, the Philadelphia Orchard Project (POP) is a non-profit organization that grows orchards in partnership with local communities [15–17]. The POP assists in orchard design, provides plant materials, organizes training workshops for orchard care, and holds events. Their community partners are responsible for taking care of the orchards and distribution of the harvest. In addition to expanding local food production, such a community-based approach contributes to fostering psychological resilience and strengthening social cohesion by providing opportunities for participants to connect [17].

However, planting edible trees in public spaces may pose potential environmental health risks and safety hazards for urban residents. First, the presence of fruit trees may attract wildlife, such as bears, into urban areas, leading to conflicts between humans and animals [11]. Second, edible tree species are susceptible to epidemics caused by large-scale insect species, such as apple moths [11]. Some cities banned planting fruit trees to prevent insect reproduction and spread [18]. Finally, there are concerns about the safety of consuming fruits from street trees. Studies conducted in different regions have

reported varying levels of lead, cadmium, arsenic, and mercury in urban-grown fruits and nuts [19,20].

Nowadays, urban residents have become less reliant on edible trees as a dietary supplement and more attracted by the fruit-picking experience. As shown in a study conducted in 47 Canadian cities, municipalities are less likely to deliberately plant edible trees in public spaces, although it is still common for fruit or nut trees to be planted on private property [11]. With that being said, edible tree species are still seen as street trees in many cities, such as mango and wood pineapple trees on the roadside in Guangzhou, China, and mango trees in Singapore [21]. These species were planted along the streets in coastal cities because their characteristics fit the local climate and environment well. Their canopy branches are large to provide shade, permeable to pass rainwater, and sparse so that they are more likely to survive typhoons. In addition, trees have a long growth cycle, and those already present in the city are not constantly replaced. However, these edible trees often lose their edible value due to insufficient nutrients, pesticide use, or picking unripe fruits early for safety reasons.

*3.2. Regulating Services of Urban Trees*

Trees deliver a variety of regulating services to urban dwellers, including carbon sequestration, oxygen release, water containment, nitrogen oxide absorption, and heat mitigation. Tree species and traits, in particular, determine what kind and how much of these services are provided (Figures 1 and 2). For example, leaf area and crown size decide a tree's capability to regulate microclimate, reduce noise, and intercept rainfall. Deciduous trees usually survive better in microclimate regulation than evergreen trees because of their larger canopy width [22]. On a larger scale, the characteristics of the surrounding green space, such as vegetation structure and configuration, also influence trees' regulating services. In the following text, we presented findings for six regulating services: heat mitigation, stormwater regulation, dust retention, toxicants enrichment, carbon sequestration, and noise reduction. For each service, we first briefly introduced the biological processes that generate the service. Then, we summarized the tree traits' contribution to the service. Last, we discussed characteristics of the surrounding environment of trees that may influence the service.

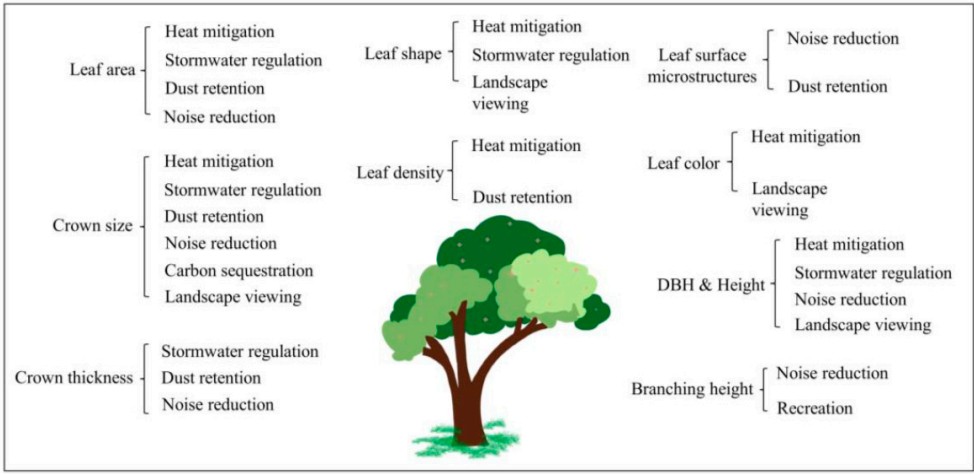

**Figure 1.** Traits affecting multiple ecosystem services.

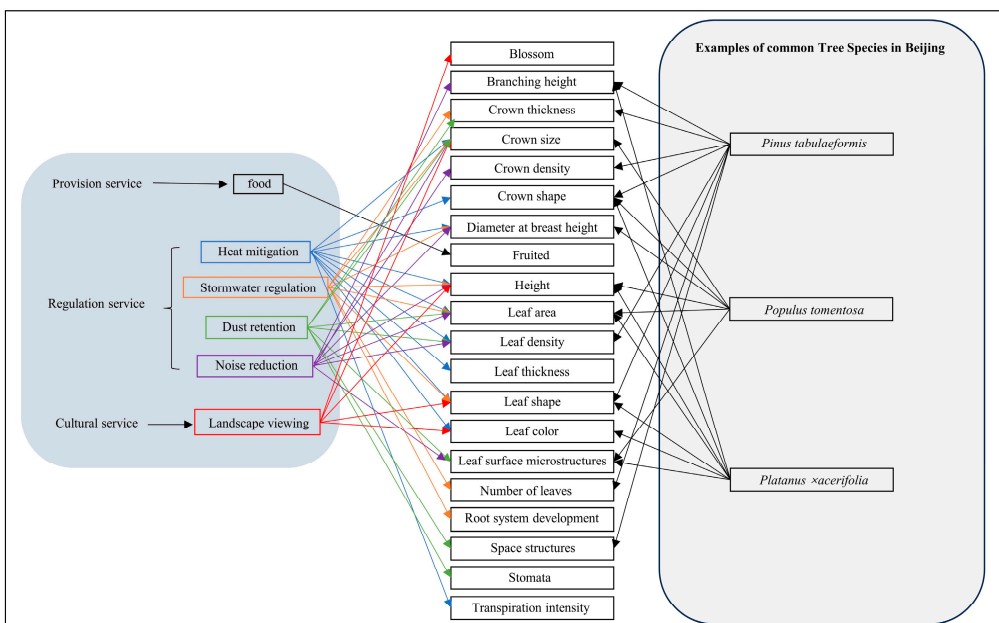

**Figure 2.** Ecosystem services and their influencing factors.

### 3.2.1. Heat Mitigation

Trees have a higher capacity to mitigate heat compared to shrubs and grasses. On clear summer days, tree-dominated areas were 0.5–1 °C cooler than shrub-dominated areas during the day and at night [23]. In addition to improving residents' thermal comfort [24], urban trees can effectively reduce the usage of air conditioning and save energy [25].

Trees cool the surrounding environment via two pathways: shading and transpiration. They provide shade to reduce the amount of heat radiated from the sun to the ground, which can reduce energy consumption by approximately 12% in residential areas and 1.3% in commercial areas [26]. Using transpiration, they absorb heat to release water and, therefore, lower air temperature. Studies measured the heat mitigation capacity of different tree species, identified relevant traits, and explored their associations based on the two pathways. Traits that are associated with heat mitigation capacity include canopy size, transpiration intensity, leaf density, and tree metabolic cycle [27]. In peak summer temperatures, a total of 332 individual trees belonging to 85 tree species were compared in the sub-canopy space with the adjacent unshaded ground. The leaf area index (LAI) and canopy size were identified as the most important factors. For example, hawthorn trees were found to provide a better cooling effect because of their higher LAI [28]. A study in Chengdu found that the weeping willow had the highest transpiration rate and resulted in approximately twice the cooling of the surrounding 1000 cubic meters of air compared to the osmanthus, which had the lowest transpiration rate.

These heat-mitigation-associated tree traits are often related to the canopy and leaf morphology. Traits related to canopy morphology are primarily linked to shade-cooling pathways. In contrast, leaf morphology traits tend to have a role in both pathways. A meta-analysis of previous studies exploring tree traits and heat mitigation effects was also conducted to investigate the magnitude of the effects of different factors [29]. It revealed that traits impacting the shade-cooling pathway had the following benefits in descending order of significance: growing size, leaf thickness, LAI, crown shape, plant functional type, wood anatomy, leaf shape, and leaf color. Similarly, the descending order in transpiration cooling is tree growth (dbh and height increment), leaf thickness, leaf shape, wood anatomy, growing size, and leaf color [29]. Based on these studies, researchers created a list of urban tree species that can effectively mitigate heat. For example, a study in Beijing, China, found that dry willow (*Salix matsudana*) has the highest cooling and humidifying capacity per

unit of leaf area, and elm (*Prunus triloba*) and plane tree (*Platanus occidentalis*) also ranked high in cooling capacity [30].

Besides canopy and leaf morphology, trees' cooling effects are influenced by other environmental factors, such as different underlayments. For example, studies have shown that the reduction in the surface temperature of grass was 3 °C per unit of LAI, while for asphalt, the reduction in surface temperature was about 6 °C [28].

### 3.2.2. Stormwater Regulation by Trees and Influencing Factors

Urban trees play a significant role in reducing stormwater accumulation. An individual "tree unit" (tree and the soil it occupies) could reduce 50% of surface runoff annually compared to a tarmac road [31]. A study in Hamilton, Canada, reported that street trees reduced 6.5–27% of the total precipitation [32].

Trees reduce surface runoff by intercepting rainwater and taking water from the ground using transpiration. Therefore, canopy thickness, root system development, and leaf shape all contribute to trees' capacity to reduce stormwater. The denser the branches and leaves, the higher the interception rate for different tree species. The more developed the root system, the easier it is for trees to promote rainwater infiltration in the soil [33]. In terms of leaf shape, research indicated that coniferous trees had an interception rate of 20–40%, while broadleaf trees' interception rate was about 10–20% [34].

The capacity for water storage within a tree's canopy is a critical factor in determining the amount of rainfall that can effectively penetrate and be absorbed. According to existing studies, the most influential tree traits in explaining rainfall interception were canopy cover and leaf area index (LAI) [35]. For example, evergreen trees possess a greater number of leaves and, therefore, exhibit lengthier leaf retention, which makes them especially vital in areas experiencing winter precipitation [36]. Among the models that estimate rainfall interception, canopy cover and LAI were the most commonly used metrics [34]. Some studies used LAI to simulate rainfall interception of individual tree canopies in urban areas [37]. However, other studies did not find a significant impact on seasonal variation of throughfall volume caused by LAI, which varied substantially in different seasons [34]. Furthermore, some studies showed that diameter at breast height and tree height were negatively correlated with tree stem flow [38].

Additionally, environmental factors such as precipitation duration and interval, ground cover, greenfield configuration [39], soil properties [40], ambient humidity, raindrop diameter, and wind speed can also influence the trees' retention of rainwater. The rainfall interception by trees increases with the intensity of rainfall until it converges to a constant. The interception also increases with the decrease in raindrop diameter, while the effect of wind cannot be generalized as it may have a mixed effect on the evaporation of trees and the reduction in interception.

### 3.2.3. Dust Retention Effect of Trees and Influencing Factors

Urban air pollution poses severe threats to human health and productivity [41]. Urban trees act as a barrier and effectively absorb atmospheric particulate matter. This section summarized studies focusing on dust retention rates of common urban tree species and the underlying factors contributing to these differences.

A study on the dust retention effects of different green space structures showed that trees and shrubs have stronger aggregation capacity than grass and are more conducive to particulate matter deposition. Studies have shown that the concentration of atmospheric dust decreases by 75% when passing through a forested area compared to an unvegetated residential area [42]. The dust from open pit coal mines was reduced by over 50% after passing through a 15-m-wide strip of birch forest [42]. Dust retention measurements conducted in Beijing revealed that the dust retention rate in mature woodlands reached up to 61.1% in summer and about 20% in winter.

Trees retain air pollutants or hinder their diffusion via two primary mechanisms: adsorbing toxic substances via the leaves and metabolizing certain toxic substances after

they are decomposed and converted into non-toxic substances in the tree's body. The dust retention capacity of trees is mainly influenced by leaf traits, canopy structure, branch and leaf density, and leaf inclination [43]. Studies focusing on tree traits have found that trees with a large leaf area and high leaf density have better dust retention effects. For example, a study conducted in Beijing discovered that species such as downy ash (*Fraxinus tomentosa*), spruce (*Picea asperata*), white pine (*Pinus bungeana*), lateral cypress (*Platycladus orientalis*), and cypress (*Juniperus sabina*) had a strong ability to adsorb and deter $PM_{2.5}$ [43]. Furthermore, comparative research showed that coniferous trees had a higher dust retention capacity than broad-leaved trees [44]. Additionally, species with hairy and deeply folded leaves exhibit better dust retention effects. For example, studies conducted on different tree species found that water cotoneaster (*Cotoneaster multiflorus*), hairy cherry (*Prunus tomentosa*), and lonicera (*Lonicera japonica*) exhibited stronger dust retention capacity due to grooved and downy leaf surface microstructures [45]. Researchers also found that torch trees (*Rhus typhina*) often have their stomata closed, resulting in a low dust retention capacity [46]. At the canopy level, trees with larger canopies and finer structures have a higher dust retention capacity.

Temperature, maintenance, irrigation, precipitation, wind speed, and environmental pollution levels all significantly affect the dust retention of urban trees. High temperature and wind speed accelerate the flow of pollutants, which reduces their adsorption and deposition by trees [47]. In addition, high levels of environmental pollution can quickly saturate the dust retention effect of trees and weaken the dust retention effect [48].

### 3.2.4. Toxicants Enrichment Effect of Trees and Influencing Factors

The rapid industrialization has led to the excessive release of heavy metals into the environment, causing significant global concern. A notable advantage of using plants for soil pollution treatment is the ability to implement in situ within projects, offering economic efficiency and avoiding secondary pollution, aligning with the overarching goal of human sustainable development. Almost 500 species of plants highly enriched with heavy metals have been identified, predominantly in metal mining regions, displaying innate tolerance to elevated heavy metal concentrations; however, the majority of these species are herbaceous plants [49]. Trees can compensate for the relatively low enrichment levels of super-enriched plants, effectively addressing urban soil pollution due to their rapid growth, substantial biomass, stable root systems, extended lifespan, pruning resistance, and ease of propagation.

Numerous studies have investigated the adsorption effects of heavy metal ions by specific tree species with distinctive structures. For instance, eucalyptus (*Eucalyptus robusta*), sycamore (*Firmiana simplex*), and acacia trees (*Acacia farnesiana*) have demonstrated greater effectiveness than other plants in the transfer of toxic metals from the soil [50]. Limited research has comprehensively examined the factors contributing to soil enrichment with heavy metals by trees. It is widely acknowledged that metal uptake by trees is influenced by numerous factors, such as fine root structure, soil texture, pH, organic matter content, and various other parameters [51,52].

### 3.2.5. Carbon Sequestration Effect of Trees and Influencing Factors

The rapid rise in atmospheric carbon constitutes a primary driver of climate change. Two strategies can mitigate atmospheric carbon concentrations: reducing carbon emissions and enhancing carbon fixation. Trees play a crucial role in mitigating the rise of atmospheric carbon dioxide by capturing, fixing, and storing it both above and below the ground [53].

The swift growth of trees in young secondary and plantation forests leads to substantial carbon absorption, with the majority stored in various tree components, including stems, roots, leaves, and branches. Carbon sequestration rates hinge on plant growth, individual tree species traits, wood density, and prevailing growing conditions [54]. The study established a relationship between soil carbon concentration and tree roots [53], specifically demonstrating that soil carbon concentration rose with increasing tree produc-

tivity, root diameter, and specific root length (SRL) while decreasing with increasing C:N content [55]. Elevated SRL corresponds to increased fine root density, leading to heightened root carbon exudation and root turnover [56]. Moreover, mycorrhizal associations of tree roots, including those with clumping fungi and ectomycorrhizal fungi, constitute pivotal factors in soil carbon sequestration [57].

Furthermore, empirical studies have identified noteworthy positive relationships between canopy area, diameter at breast height, tree height, and carbon sequestration. Similarly, soil carbon storage exhibited positive associations with photosynthesis rate, chlorophyll a, chlorophyll b, and carotenoid content. Additionally, diverse regression models revealed that water use efficiency and stomatal conductance serve as robust predictors of soil carbon stocks in bristlecone pine forests [58].

From an environmental perspective, factors such as elevation and precipitation exert indirect influences on carbon sequestration by affecting tree growth.

3.2.6. Noise Reduction Effect of Trees and Influencing Factors

Sound attenuation occurs when sound waves pass through dense plant clumps higher than 1m because of plant blockage. Trees are often planted along roads to reduce traffic noise. According to the capillary absorption coefficient formula (Formula (1)), the coefficient of absorption for capillaries exhibits an inverse relationship with the radius of the tube and a direct relationship with the square root of the frequency.

$$\alpha = \frac{2}{ac_0}\sqrt{\frac{\gamma\omega\eta}{\rho_0}}, \tag{1}$$

In this equation, $\alpha$ is the sound absorption coefficient, $a$ is the radius of the tube, and $\frac{\gamma\omega\eta}{\rho_0}$ is the frequency [59].

The finer the capillary and the higher the noise frequency, the greater the absorption coefficient is. In terms of tree traits, previous studies have suggested that bushy branches, large leaves, and hairy leaf surfaces all constitute fine capillaries and, therefore, have better sound insulation effects on noise [60].

Studies indicated that the amount of noise attenuation in forest belts within the 500–2000 Hz frequency range was largely contingent upon the quantity and structure of leaves. The sound absorption of trees was proportional to the number of their leaves [61]. However, frequencies exceeding 2000 Hz are primarily influenced by factors such as tree height and external morphology. For example, broad-leaved trees reduce noise better than coniferous trees, indicating leaf area was a critical indicator [61].

Additionally, researchers have conducted extensive research on the noise abatement effect of green belts. A study in Shanxi, China, revealed a highly significant positive correlation between noise attenuation in green belts and trees' diameter at breast height [62]. The branching height of plants was also identified as one of the most important factors affecting the noise attenuation effect. Shrubs with lower branching point heights and denser branches and leaves tend to have better scattering effects on sound waves and can achieve a better noise attenuation effect [63].

It is worth noting that some scholars suggested that for noise reduction benefits, the length, width, height, density, and arrangement of green belts played more important roles than the morphological characteristics of the plants themselves [64]. For example, research indicated that green belts wide 16–70 m can effectively reduce noise [65]. In addition, the arrangement of trees and shrubs in the green belt also affects its noise reduction effect. It is generally agreed that a mixed patch of trees and shrubs reduces noise more effectively than that of a single shrub or tree patch [66]. Additionally, the higher the density of trees and shrubs, the better the noise reduction effect [66]. Planting green belts near the noise source is more effective at noise reduction than planting green belts near the protected object [67]. Additionally, the ground covered by branches, leaves, and dead branches is effective at absorbing low and medium-frequency noise [68]. With that being said, the foliage in cities

is typically cleaned regularly to eliminate fire hazards and maintain a tidy environment, which can eliminate this noise reduction pathway.

### *3.3. Cultural Services of Urban Trees*

As an important component of urban green space, trees play an essential role in providing cultural service. Urban trees contribute to a wide array of cultural services, encompassing landscaping enhancement, alleviating physical and mental stress among residents, augmenting property value, and so on. Urban trees provide a wide variety of cultural services, which has attracted considerable research interest. The reason the association between cultural service and traits took up much less space in this article is that compared with regulating services, the realization of cultural service is a much more complicated process involving many other aspects, in which tree traits played a relatively small role. Therefore, when delving into the association between cultural services and tree traits, we primarily emphasize their ornamental significance. Many traits were studied for their contribution to enhancing visitors' recreational experiences. Some were agreed as important contributors, such as colored foliage and flowers. Others were reported to have mixed results, such as heights and crown size.

Researchers came to some agreements on preferences of shapes and colors. People have different emotional experiences when looking at trees with different shapes. Healthy trees were generally preferred over those that had been damaged or were withered [69]. People generally preferred street trees with conical or spreading shapes [70]. Leaf color and flowers are important factors when it comes to aesthetic evaluation. Two or three colors are usually considered ideal since too many colors seem chaotic, while a single color is monotonous [71]. Colored foliage species enrich the seasonal changes of parks, while the combination of different flowering species prolongs the park's enjoyment period.

However, studies examined public preferences for other tree traits, such as height, crown size, density, shape, and leaf size, came to inconsistent results. For example, some studies found that tall trees were favored due to their ability to offer better views [72], while others found that preferences for tree height varied [73]. Furthermore, trees with large and dense crowns were generally preferred since they indicated good health and contributed to overall landscape aesthetic quality [74]. With that being said, some studies found that participants did not show any preference for particular canopy densities [75]. Results for leaf width were also inconsistent. Some studies revealed a preference for large leaves during summer [69], while others found that people like small leaves because they look more delicate [70].

## 4. Discussion and Conclusions

Besides these benefits trees can provide, trees also have adverse effects on humans, which are sometimes referred to as disservice. Related research has focused on aesthetic issues and environmentally harmful consequences, health and safety impacts, and management costs associated with ecological disturbance and risk management [76]. Aesthetics is sometimes a central motivation for residents to plant trees, but it can also be a disservice in some cases, such as leaf, branch, or fruit litter disposal [77]. Trees may interfere with or damage infrastructure, such as hiding traffic signs or roots raising sidewalks. In extreme weather, hurricanes, for example, falling branches often cause further property damage [78]. In addition, sensitization and biological invasions are disservices that cannot be ignored. For example, some trees are allergenic because of their original pollen, fruit hairs, or fruits [79]. Relevant studies include pollen dispersal patterns, the flowering and fruiting periods of allergenic species, and the generalization of urban allergenic tree lists [80]. Other adverse effects may include bio invasions caused by introducing exotic species for ornamental purposes, as well as increased water use by trees, especially in cities where water is scarce [76]. Disservice produced by trees is often related to their size, species, and location [81], and a growing body of literature on tree risk management focuses on assessing and mitigating these damages. While this review primarily focused

on the positive side of their benefits, the disservice of urban trees is an important issue in both research and practice.

This paper summarized the association between tree traits and provision, regulation, and cultural services (Tables 1 and 2). It is clear that tree trait mediates trees' ecosystem services. We thereby argue that a trait-service framework is needed to access trees' ecosystem service supply. It is worth noting that restoration ecology has adopted a similar "trait-benefit" approach as the theoretical basis to identify a combination of species that can accomplish the specific restorative task (such as absorbing heavy metal in soil) and survive the target environment in the meanwhile, which is often polluted [82,83]. Furthermore, restoration ecologists developed "trait-benefit" databases, namely BiolFlor and TRY, to document individual tissues' morphological, anatomical, or physiological traits based on peer-reviewed publications [84].

**Table 1.** Urban trees' regulation services and their influencing factors.

| Ecosystem Service | | | Influencing Factor | |
|---|---|---|---|---|
| | | | **Tree** | **Environment** |
| Regulation service | Heat mitigation | Shade to achieve heat mitigation | Duration of metabolism | Underlayment |
| | | | Leaf area<br>Leaf density<br>Leaf thickness<br>Leaf shape<br>Leaf color<br>Crown size<br>Crown shape<br>Diameter at breast height<br>Tree height | |
| | | Transpiration to achieve heat mitigation | Transpiration intensity | Temperature |
| | | | Crown size | |
| | Stormwater regulation | | Leaf shape | Duration and interval of precipitation |
| | | | Leaf area<br>Number of leaves<br>Crown size<br>Crown thickness<br>Diameter at breast height<br>Tree height<br>Root system development | Humidity<br>Diameter of raindrop<br>Wind |
| | Dust retention | | Leaf area<br>Leaf density<br>Leaf surface microstructures | Temperature<br>Wind speed<br>Irrigation and precipitation |
| | | | Stomata | Level of environmental pollution |
| | Toxicants enrichment | | Space structure (crown density and leaf inclination) | Maintenance status |
| | | | Crown size<br>Crown thickness<br>Fine root structure<br>Soil texture<br>pH<br>Organic matter content | |

**Table 1.** *Cont.*

| Ecosystem Service | Influencing Factor | |
| --- | --- | --- |
| | **Tree** | **Environment** |
| Carbon sequestration | Wood density<br>Root diameter<br>Specific root length<br>C:N content<br>Mycorrhizal<br>Crown size<br>Diameter at breast height<br>Tree height<br>Photosynthesis rate<br>Chlorophyll a<br>Chlorophyll b<br>Carotenoid content<br>Water use efficiency<br>Stomatal conductance<br>Leaf area | Growing conditions<br>Elevation<br>Precipitation |
| Noise reduction | Crown size<br>Bushy branches<br>Leaf surface microstructures<br>Tree height<br>Diameter at breast height<br>Branching height | Remaining dead branches and leaves |

**Table 2.** Urban trees' cultural services and their influencing factors.

| Ecosystem Service | | Influencing Factor | |
| --- | --- | --- | --- |
| | | **Tree** | **Environment** |
| Cultural service | Landscape Viewing | Leaf shape<br>Leaf color<br>Blossom<br>Height<br>Crown size<br>Fruit | Water *<br>Terrain<br>Animal<br>Landscape Design<br>Soil pH<br>Rainfall |
| | Recreation | Under branch height<br>Allergenicity | Temperature<br>Recreation facilities<br>Accessibility<br>Route Planning |
| | Research and Education | | Plant species in the park<br>Observatory<br>Exhibition sign |
| | Cultural Heritage | Old and valuable trees | |

* On a slightly smaller scale, it also involves tree species matching and space structure.

Furthermore, every tree species essentially represents a bundle of several traits, with each trait making contributions to multiple services, which is illustrated in Figure 1 using three commonly seen tree species in Beijing, China. Therefore, there is no one-on-one correspondence between traits and ecosystem service. Instead, picking any species indicates a variety of traits, bringing a set of services. The synergistic or trade-off among them needs to be further studied (Figure 3). Potential species for urban tree planting vary according to climate and cultural context, so a place-based effort would be appropriate to generate a local inventory of tree species, traits, and their service, providing scientific support for tree-planting practice from an ecosystem service perspective. One example of application is to generate recommendations of tree species that should be planted in particular areas in

a city, such as pedestrian routes, residential areas, parks, or traffic zones, where trees were expected to provide different combinations of services. Trees in residential areas are usually appreciated for aesthetic values, and allergenic species should be avoided. In contrast, trees in traffic zones were planted mostly to reduce pollutants and noise. Researchers can thus explore the weighting of different traits in a more detailed and refined way and have the opportunity to amalgamate the requirements of distinct urban zones. This enables the identification of suitable combinations of tree traits tailored to diverse contexts. A place-based inventory based on the "trait-service" framework would help city planners pick appropriate species in different locations. This review should serve as a starting point to develop a "trait-service" framework for ecosystem service research. With the help of such a framework, future research should generate actionable knowledge for practitioners to identify potential tree species for selection according to desired services.

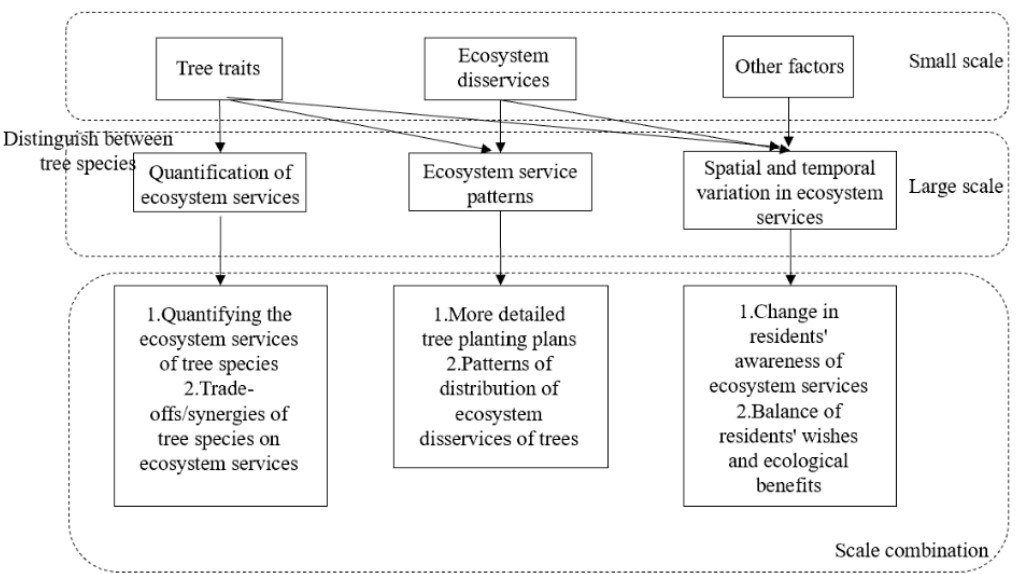

**Figure 3.** Research directions for exploring ecosystem services based on tree traits.

**Author Contributions:** Conceptualization, D.L. and G.H.; validation, D.L. and G.H.; investigation, D.L.; writing—original draft preparation, D.L.; writing—review and editing, D.L. and G.H.; supervision, G.H. All authors have read and agreed to the published version of the manuscript.

**Funding:** This research received no external funding.

**Data Availability Statement:** No new data were created or analyzed in this study. Data sharing is not applicable to this article.

**Conflicts of Interest:** The authors declare no conflict of interest.

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
