# Peer review of "Influence of Urban Tree Traits on Their Ecosystem Services: A Literature Review"

_land, doi:10.3390/land12091699_

Round 1

Reviewer 1 Report

Comments to the „Influence of urban tree traits on their ecosystem services: A review” publication

This study examines the extent to which peer-reviewed articles address the services provided by trees rather than the whole green space ecosystem.

Below are my comments

-          Chapter 2 does not tell us much new; fruit trees planted in masses in the past should not be planted today, because it is well known that the polluted air and soil of a polluted urban environment has a negative effect on fruit yield. In addition, when the fruit ripens and falls, it is more difficult to maintain.

-          Between lines 216-218 and elsewhere in the work, the scientific names of species are used instead of common names. The binomial Latin name is obvious to everyone, while the fact that, for example, torch tree has no idea what species.

-          In chapter 3.4, specify the operators in the formula, i.e. put the abbreviations and acronyms in the text explanation.

-          When it comes to cultural services, it is also worthwhile to collect the role and "services" of the green environment in recreational, physical, and mental health. In this respect, they are missing from the table and the text. Also missing are the beneficial changes associated with the green environment, such as changes in public amenity values, and property values. If the cultural aspects are brought in, they should not be left out.

-          There is also a lack of substance in the mention of negative effects, even if it is written that this is not the focus. It does not mention anything other than allergies and invasion possibilities.

-          Table 1: in the case of transpiration not only transpiration intensity but also canopy area is an important factor;

other aspects are also missing for stormwater regulation, including, but not limited to, ground cover, tree location, planting methods, innovation, etc.;

for dust retention, density, quality of maintenance, and whether irrigation is available (several scientific publications have been written on this).

-          Table 2: autumn foliage colour is influenced not only by terrain but also by pH, rainfall, and temperature.

-          Figure 1: needs to be added;

the examples are not very clear; e.g. why is the foliage colour not important for Populus tomentosa, etc. etc.;

for landscape I think space structure is also important.

Overall, I missed the article on the context, and the weighting of the different features in order to maximise the number of services that can be provided if designers choose the right tree species and gardeners plant the right trees. If we focus on the literature review, there is also scope for expanding the material there, within the scope.

Reviewer 2 Report

The effect of urban tree features on their ecosystem services is discussed in this review. Overall, the review is technically sound; however, there are a few queries and remarks that need explanation.

Not only the properties of the leaves, but also the characteristics of the tree crowns (crown size, crown thickness, crown density, etc.) affect the capacity of trees to retain dust. This point requires additional debate.

Figure 1. Only three species of woody plants (common trees…) were employed in the greening of Beijing's city?

Very essential characteristics of urban trees for ecosystem services are just briefly mentioned in the review's text; however, they are not examined in detail or compiled in tables and figures. This is the capacity of woody plants to lower the number of pollutants in the air, mostly by the uptake of toxic gases (nitrogen oxides, sulphur, volatile hydrocarbons, and so on) by leaves. The second capability involves toxicants being absorbed by tree roots from the soil and then being deposited into the plant mass.

The role of wood plants in carbon sequestration was not included in the review.

The review cannot be deemed comprehensive without addressing how these two key roles of woody plants in ecosystem services. Therefore, the review has to be significantly revised.

Reviewer 3 Report

I have gone through the manuscript entitled: “Influence of urban tree traits on their ecosystem services: A review” and found it interesting. However there is a deep modification required towards improving its readability:

I suggest changing a little bit the title of the article instead of " a review" to " A literature review"

Line 15 "Studies focusing on 15 ecosystem services often consider green space as a whole, and some recognized the difference between trees and grass"

Not "recognize" but "distinguish"

Indicate what is the purpose of the study, what are the research questions? What do the authors want to achieve with this research?

Line 49- to name a few does not fit in this sentence or needs to be corrected grammatically.

Line 58-66 describes the methodology. I suggest distinguishing this part in a separate subsection methods and methodology and at the same time enriching the Introduction part with an analysis of the literature on the subject in terms of other similar studies as well as an indication of the research gap.  Then, unfortunately, the layout of the article should be changed and the parts about tree services should be included in a subsection that could be called generally ecosystem services provided by trees, or else.

I suggest adding a graphic to the article that would capture the types of trees and ecosystem services. This would be an added value to the work that could clearly summarize the entire article in one place. You could also add a couple of graphics that would capture the names and characteristics of trees with their ecosystem services.

It would be good to add a recommendation as to which species of trees should be planted in particular areas of cities ( near roads, pedestrian routes, residential areas, parks, etc.) or consider this suggestion for other studies.

No comments

Round 2

Reviewer 1 Report

I accepted it, please improve the official Latin names as:

Cerasus tomentosa is Prunus tomentosa; Sabina chinensis is Juniperus sabina

Reviewer 2 Report

All the issues I cared about were addressed by the authors. I believe the manuscript can be accepted for publication.
